# Micro-Architectural Investigation of Teleost Fish Rib Inducing Pliant Mechanical Property

**DOI:** 10.3390/ma13225099

**Published:** 2020-11-12

**Authors:** Yu Yang Jiao, Masahiro Okada, Emilio Satoshi Hara, Shi Chao Xie, Noriyuki Nagaoka, Takayoshi Nakano, Takuya Matsumoto

**Affiliations:** 1Department of Biomaterials, Graduate School of Medicine, Dentistry and Pharmaceutical Sciences, Okayama University, 2-5-1 Shikata-cho, Kita-ku, Okayama 700-8558, Japan; phue18vu@s.okayama-u.ac.jp (Y.Y.J.); m_okada@cc.okayama-u.ac.jp (M.O.); gmd421209@s.okayama-u.ac.jp (E.S.H.); shasecho@yahoo.co.jp (S.C.X.); 2Advanced Research Center for Oral and Craniofacial Sciences, Graduate School of Medicine, Dentistry and Pharmaceutical Sciences, Okayama University, 2-5-1 Shikata-cho, Kita-ku, Okayama 700-8558, Japan; nagaoka@okayama-u.ac.jp; 3Division of Materials and Manufacturing Science, Graduate School of Engineering, Osaka University, Suita 565-0871, Japan; nakano@mat.eng.osaka-u.ac.jp

**Keywords:** bone-like material, mechanical property, orientation, layered structure

## Abstract

Despite the fact that various reports have been discussing bone tissue regeneration, precise bone tissue manipulation, such as controlling the physical properties of the regenerated bone tissue, still remains a big challenge. Here, we focused on the teleost fish ribs showing flexible and tough mechanical properties to obtain a deeper insight into the structural and functional features of bone tissue from different species, which would be valuable for the superior design of bone-mimicking materials. Herein, we examined their compositions, microstructure, histology, and mechanical properties. The first rib of Carassius langsdorfii showed a higher Young’s modulus with a small region of chondrocyte clusters compared with other smaller ribs. In addition, highly oriented collagen fibers and osteocytes were observed in the first rib, indicating that the longest first rib would be more mature. Moreover, the layer-by-layer structure of the oriented bone collagen was observed in each rib. These microarchitectural and compositional findings of fish rib bone would give one the useful idea to reproduce such a highly flexible rib bone-like material.

## 1. Introduction

Bone tissue engineering has received considerable attention for the past several decades [1,2,3]. In this context, attempts to create bone tissue using biological and artificial materials have been extensively studied. For example, the use of bone morphogenetic protein (BMP) or basic fibroblast growth factor (bFGF) with carrier matrices such as decalcified bone [4], collagen, and non-collagenous proteins (NCPs) are recognized as valuable osteoinductive materials [5,6]. Bone regeneration using cells such as mesenchymal stem cells having high osteoinductive ability has also been considerably studied [7]. In addition, a variety of methods for the fabrication of bone engineering scaffolds have been introduced to provide a better environment for cells and regenerating tissues [8,9,10,11]. Through these efforts, clinicians are now able to regenerate the bone tissue to some extent. Therefore, the research interests related to bone tissue regeneration are now shifting to more precise control of the regenerated bone.

In recent years, the possibility of controlling the physical properties of the generated tissue has been increasing from the viewpoint of biomimetics [12]. One possibility is to mimic the structure of bone tissue. Histological sectioning is a general method for investigating tissue structure in biology. In addition to this, combining ultrastructural observation and the mechanical/chemical evaluation of biological tissue would be an effective and original approach in the material design for tissue engineering. In other words, an engineering and material science perspective may be valuable for this purpose. For example, our group indicated the importance of a cell membrane fragment for the rapid mineralization through the investigation of femur epiphysis bone development [13,14,15]. Extensive research studies have been done on the mechanical properties and structure of bone at various taxonomic levels, including mammals, amphibians, and birds. For example, research has been conducted on how the structure and composition of river turtle and crocodile bones support the body [16]. Similarly, investigation of the structure and properties of fish bones indicated that fish bones can be classified into cellular and acellular bones, and there were significant differences in the mechanical properties and mineral content of each bone [17]. However, we still do not know why the fish bone shows such soft and tough mechanical properties. In this study, we focused on the ribs of Carassius langsdorfii and examined their compositions, microstructure, histology, and mechanical properties through an engineering perspective.

## 2. Materials and Methods

### 2.1. Sample Preparation

Adult Carassius langsdorfii with an average length of 70 ± 8 mm were used for the studies. Both sides of the first (rib1), fifth (rib5), and ninth (rib9) ribs were isolated (Figure 1A). Excess connective tissue of the isolated bones was removed under the dissection microscope using tweezers. The samples were stored in 0.5% saline solution at 4 °C for no more than 6 h before use. In addition, to investigate the effect of water content on the properties of bone tissue, the samples were heated at different temperatures from 50 to 110 °C.

### 2.2. Mechanical Test

The samples, previously immersed in 0.5% saline solution, were placed onto a three-point bending jig of a mechanical tester immediately before the flexural tests. To ensure consistency in the measurements, the center of the isolated intact ribs (rib 1, rib 5, and rib 9) was placed in the center of the three-point bending jig (Figure 1B). The distance between the two ends supporting the bones was 10 mm. The test was measured using a table-top mechanical tester (EZ-SX 500N, Shimadzu Corp., Kyoto, Japan) with a loading method based on a high-precision constant-speed strain measurement using a backlash-free ball screw drive. The crosshead speed for the measurement was set to 0.1 mm/s. Fifteen samples of each rib were used in the measurements. The resulting stress–strain curve was used to calculate the Young’s modulus of the rib material.
(1)$$E=\frac{S\cdot L^{3}}{48I}$$
(2)$$I=\frac{b\cdot d^{3}}{12}$$
(3)$$S=\frac{\Delta F}{\Delta L}$$

In this formula, E (N/mm^2^) is the Young’s modulus of the bone material in the direction coinciding with the long axis of the rib. S (N/mm) is the slope of the linear part of the force displacement curve. L (mm) is the span distance between the stationary supports, and I (mm^4^) is the moment of inertia of each rib, where b is the rib width (mm) and d is the rib depth (mm). In this experiment, the cross-section of the rib bone was assumed to be rectangular.

### 2.3. Characterization of Fish Rib Bone

The isolated ribs were weighed with an analytical scale (±0.05 mg) to determine the lipid-free wet mass. Then, the samples were placed in a ceramic cup and heated in an oven (Koyo Thermo Systems Corp., Nara, Japan) to 100 °C for 3 h to remove all unbound water. After heating, the samples were immediately weighed to determine their dry mass (organic and mineral content). Next, samples were heated in an oven at 500 °C for 10 h to eliminate all organic material. The remaining inorganic material was weighed immediately to measure the dry bone content (%) of each sample. Water (wet mass—dry mass), organic (dry mass—ash mass) and inorganic contents were also calculated by dividing by wet mass to determine a percentage.

For analysis of the crystallinity of the fish ribs, the samples, previously fixed with 4% paraformaldehyde (PFA) for 24 h, were used for thin-film X-ray diffraction (XRD) measurements (RINT2500HF; Rigaku Corp., Tokyo, Japan) at an incidence angle of 1° using Cu-Kα (1.54 Å) irradiation at 40 kV and 200 mA. The XRD measurements were conducted from 15° to 45° at a scan speed of 1° min^−1^. The degree of preferential orientation of the apatite c-axis was evaluated by the relative intensity of the diffraction peak: (002), (211), and (310) respectively in the XRD profile [18].

The attenuated total reflectance Fourier transform infrared spectroscopy (ATR-FTIR, JISC6802, Shimadzu Corp.) spectra were recorded after pressing the fish bone samples directly on a ZnSe Prism (MIRacle 10; Shimadzu Corp.). The result was analyzed by spectrum analysis software (LabSolutions IR version 2.13; Shimadzu Corp.).

### 2.4. Microstructure Observation

For inorganic material observation, freshly isolated rib samples were maintained in 10% NaClO solution for 3 days. The NaClO solution was changed daily. For organic material observation, the PFA-fixed samples were decalcified with ethylenediaminetetraacetic acid (EDTA) solution for 2 weeks. Finally, the samples were freeze-dried, mounted, and coated with osmium for SEM observation (JSM-6701F; JEOL Ltd., Tokyo, Japan). The decalcified samples were embedded in paraffin. The paraffin thin sections were stained with hematoxylin and eosin (HE). The optical microscope with a digital camera was used to obtain images of the stained samples (CKX41N-31PHP, Olympus Corp., Tokyo, Japan).

### 2.5. Statistical Analysis

The data were averaged and represented as mean ± standard deviation (SD). Statistical comparisons between the two means were performed using a two-tailed unpaired Student’s t-test followed by a F-test for homoscedasticity. *p* < 0.05 was considered as significant.

## 3. Results

### 3.1. Mechanical Property of Rib Bone

Fish bones have soft and strong mechanical properties [19]. In order to understand the reason for this, we investigated the microarchitecture and constituents of the fish bone. Especially, in this study, focusing on the Carassius langsdorfii, the large size rib1, the medium size rib5, and the small size rib9 were extracted and compared. When examining the mechanical properties, each rib showed a typical stress and strain profile. In terms of Young’s modulus, rib1 showed the highest, and the value became lower with the increase the rib numbers. In order to examine the effect of water and organics on this suppleness, the extracted bone was heated to remove the water, and the mechanical properties were examined. Increased Young’s modulus of the heated bone tissue was observed (Figure 1).

### 3.2. Characterization of Fish Rib Bone

Next we examined the contents of water, organic components, and inorganic components by measuring the weight in the drying process. In any ribs, the water content was approximately 5%, the organic components were 35–40%, and the inorganic components were less than 60%. Among these, no significant difference was observed. As a result of XRD analysis, all ribs showed apatite-based diffraction peaks. Rib9 showed lower crystallinity than other rib1 and rib5 according to the sharpness of peaks. As a result of examining each peak based on the results, it was confirmed that the crystal had a high orientation in the long axis direction [20]. In the FTIR study of the components, peaks of carbonate were also observed in addition to peaks of phosphate (Figure 2).

### 3.3. Microstructure Observation

SEM images of decalcified bone showed that the collagen fibers were highly oriented toward the long axis of the ribs. In the remaining minerals after removing the soft tissue, the localization of apatite crystals was consistent with the presence of fibrous collagen. In addition, it was revealed that each crystal having a flake-like morphology showed an orientation similar to that of collagen. The isolated rib bone has a flat shape in the external and internal directions. Interestingly, it was understood that this flat shape was formed by stacking lamellar collagen fiber layers (Figure 3).

When the internal structure of the rib bone tissue was examined, a non-calcified lumen structure was observed inside each rib. In addition, when this tissue was observed in an enlarged image, it was found that the space was occupied by hypertrophic chondrocytes rather than bone marrow tissue. It was also found that the proportion of this chondrocyte area in the center of the ribs decreased from rib9 to rib1 (Figure 4).

To understand the mechanism of this orientation, the orientation of osteocytes constituting the bone matrix was examined from the HE-stained images. Interestingly, osteocytes existing on the outer region of the bone matrix showed a better cellular alignment than those in the inner region. The results were similar regardless of the rib number (Figure 5).

## 4. Discussion

In recent years, various soluble factors, cells, and biomaterials have been applied to achieve bone regeneration [21]. As a result, the regeneration of bone defects of small size, such as tiny alveolar bone defects in periodontal diseases, has been widely achieved [22]. Therefore, in the future, research on bone regeneration will be advanced for more complicated purposes, such as quality control of regenerated bone. To this end, it may be useful to refer to the bone properties and structure of other animal species. Fish have less need for organ protection than the ground animals because their habitat is underwater, and they are always swimming in the water, so they may have acquired flexible bone mechanical properties. In this study, we evaluated how the suppleness of fish ribs is made in terms of structure and components.

Rib1 showed a stiffness of approximately 12 GPa in Young’s modulus. In general, the Young’s modulus of human long bones is approximately 20 GPa [23,24]. Therefore, the fish ribs are considerably softer and more supple. Moreover, this study results suggest that rib1 of Carassius langsdorfii would be a more mature bone than rib5, which in turn would be a more mature bone than rib9. The composition of this bone mainly consists of carbonate apatite and is similar to that of mammals and birds [25,26]. However, the composition of organics and water accounts for 40% of the total weight, which is characteristically different from those of terrestrial organisms. In fact, it is known that in mammals, such as humans, the composition of organics and water accounts for 30%, and in birds, it accounts for 40% [27]. One of the reasons for the suppleness of the fish bone would be that the proportions of water and organic components are high, which is in part similar to the bird bone. Therefore, to confirm this effect, the isolated rib bone was heated to reduce the water content, and then the mechanical properties were measured. As a result, it was clarified that the water content was almost eliminated by heating for 6 h, and consequently, the Young’s modulus increased remarkably with a decrease in the water content.

A structurally interesting feature is that the interior lumen of the rib is almost occupied by chondrocyte-like cells and there is no bone marrow tissue. It is known that fish has no bone marrow tissue because hematopoiesis is mostly carried out in the kidney [28]. However, in terms of the mechanical properties of bone, even though there is no bone marrow tissue, this non-calcified lumen structure may be important for maintaining the supple properties of the rib bones. In addition, all the fibers of rib tissues showed high orientation along the long axis direction of the bone from observation of the microstructure by SEM [29]. This characteristic collagen fiber orientation plays an important role in maintaining their strength [30]. More interestingly, rib bones showed a structurally layered structure. The stacking direction of this layered structure was perpendicular to the direction in which the deformation occurs due to the expansion and contraction of the fish organs. From this result, it is considered that as the body grows, the deformation of the organs also increases, and along with this, the stratification progresses so as to reinforce the resistance to the deformation. This idea is supported by the better alignment of the outer bone cells of the bone matrix than the inner bone cells. Additionally, a better alignment of osteocytes in longitudinal normal bone tissue compared to diseased immature bone has been reported previously [31]. Although further studies are needed to clarify the mechanisms involved in the cellular alignment and the layered structure formation, this specific structure may considerably relate to the flexible property of fish ribs.

## 5. Conclusions

In conclusion, we found that fish rib bones exhibit specific mechanical properties, presenting a particularly high flexibility, which was associated in part with a characteristic laminar structure and orientation of collagen fibers, as well as a high organic and water content. This information would be important in material design for making soft and strong bones.

## Figures and Tables

**Figure 1 materials-13-05099-f001:**
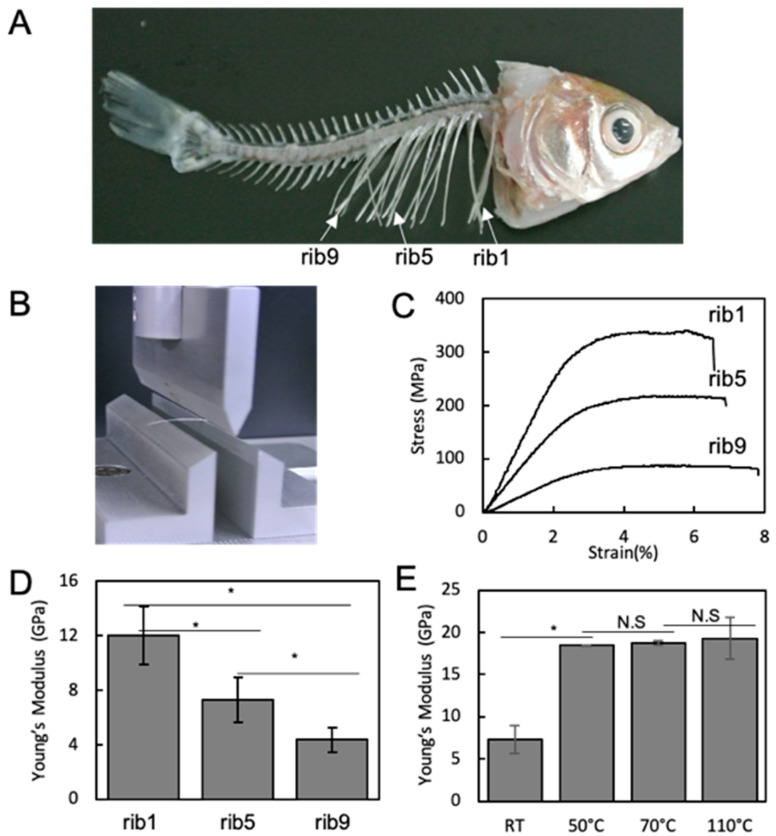
Mechanical property of Carassius langsdorfii rib bone. (**A**) Fish skeletal. Arrows indicate ribs 1, 5, and 9. (**B**) Three-point bending jig used in this study. (**C**) Stress–strain profiles of fish rib. Rib 1 has a higher strength than rib 5 and rib 9. (**D**) Young’s modulus of fish ribs. The more distal the location of the rib, the softer it becomes. (**E**) The alteration of Young’s modulus of rib 5 by heating. * indicates a significant difference (*p* < 0.05).

**Figure 2 materials-13-05099-f002:**
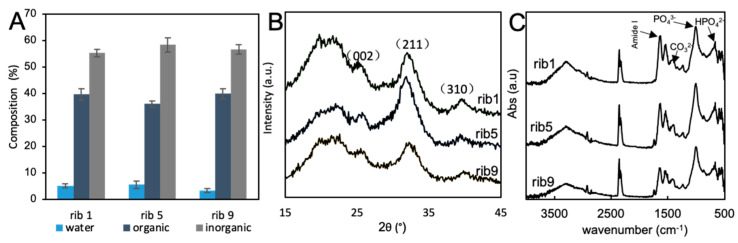
Chemical and crystallographical property of Carassius langsdorfii rib bone. (**A**) Composition of fish ribs. Minerals account for less than 60% of the total, and organic and water account for more than 40–45% of the total. (**B**) XRD profiles of fish ribs. The higher the rib number, the lower the crystallinity becomes. (**C**) Attenuated total reflectance Fourier transform infrared spectroscopy (ATR-FTIR) spectrum of fish ribs. The presence of carbonate in addition to phosphate was confirmed.

**Figure 3 materials-13-05099-f003:**
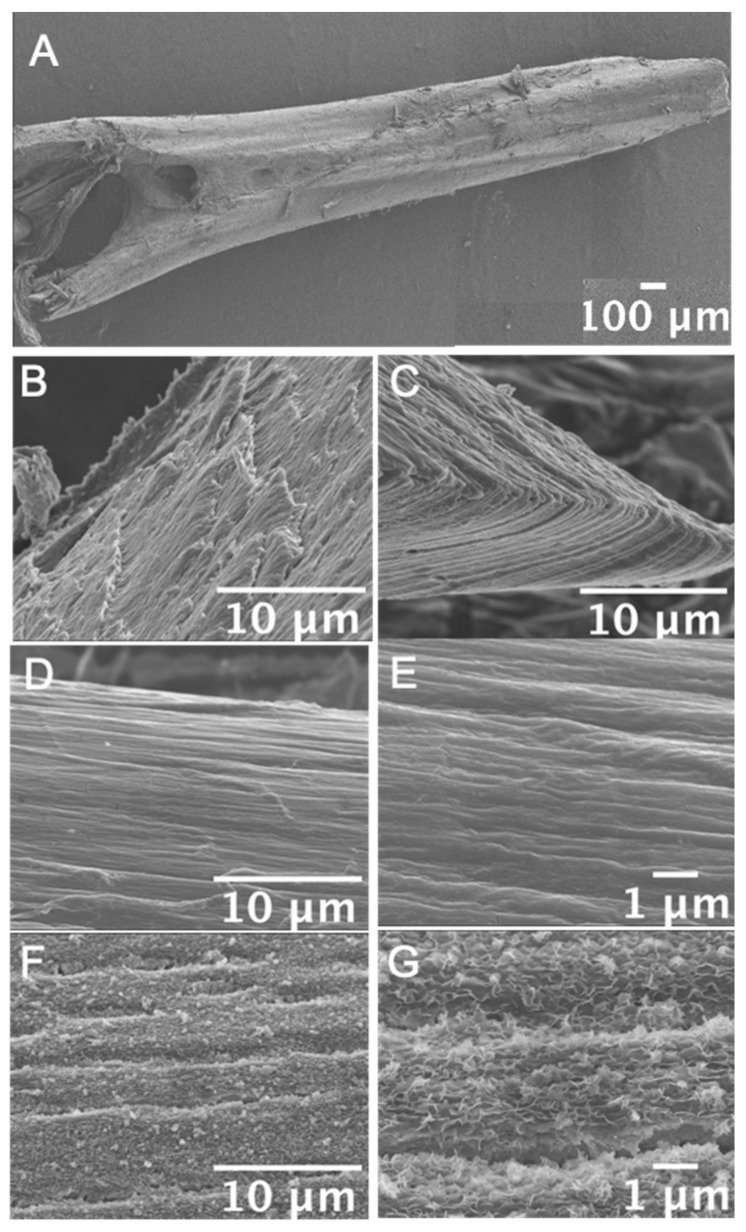
Microstructure of Carassius langsdorfii bone rib. (**A**) Low magnified SEM image of fish rib tip. Each rib shows a flat shape. (**B**,**C**) The lamellar collagen fiber layers stacked to form this flat shape. (**D**,**E**) SEM images of rib bone organic surface treated with ethylenediaminetetraacetic acid (EDTA). Collagen bundles were oriented identically to the long axis of rib bone. (**F**,**G**) SEM images of rib bone inorganic surface treated with NaClO. Small inorganic crystals were aligned identically to the organic orientations.

**Figure 4 materials-13-05099-f004:**
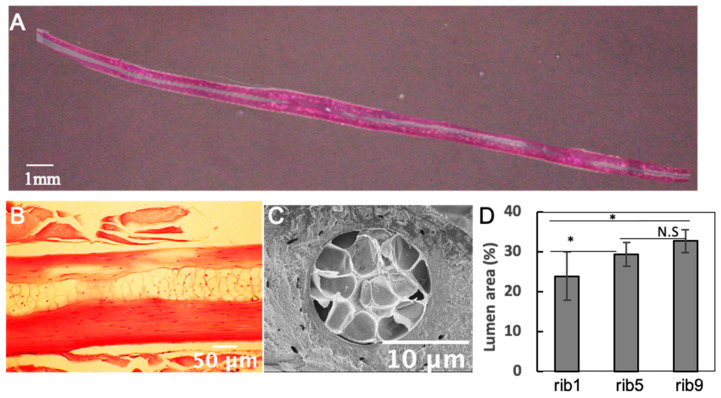
Histological observation of Carassius langsdorfii rib bone. (**A**,**B**) Hematoxylin and eosin (HE) stained images of cross-section of rib5. A cavity structure is found in the center. (**C**) SEM image of the cross-section perpendicular to the long axis of rib5. The presence of hypertrophic chondrocytes can be seen in the central part. (**D**) The larger the number of ribs, the larger the lumen observed. * indicates a significant difference (*p* < 0.05).

**Figure 5 materials-13-05099-f005:**
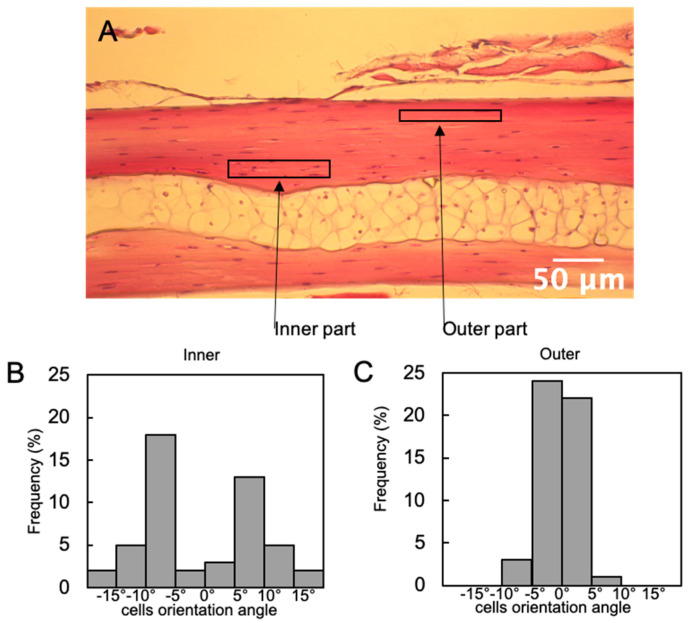
Cell alignment in Carassius langsdorfii rib bone. (**A**) Representative image of rib cross-section stained with HE. (**B**,**C**) In the cross-section of the fish rib, the outer osteocytes far from the lumen show better cell alignment than the inner osteocytes. The same result was observed in either rib.

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
