# Peer review of "Micro-Architectural Investigation of Teleost Fish Rib Inducing Pliant Mechanical Property"

_materials, 2020, doi:10.3390/ma13225099_

Round 1

Reviewer 1 Report

This paper is focused on the ribs of Carassius langsdorfii, and examined their compositions, microstructure, histology and mechanical properties through engineering perspective. In the introduction, there are many references to existing studies, but in comparison with this, it is necessary to mention in which part there is creativity and originality in examining the mechanical properties etc. through engineering perspective covered in this paper.

In 3.1, it is very difficult to measure the mechanical properties of fish bones and requires advanced technology, but details for measure method are missing.

In the discussion, there are many references to existing studies, but in comparison with this, it is necessary to mention and review main factors for maintaining flexible bone tissue properties.

In the conclusion, the logic for reaching the conclusion is very insufficient. In the case of this paper, it focuses on investigating structure and measuring mechanical properties the introduction, but for this, creativity and originality in measurement technology are required. If it is intended to be applied to other parts by measuring the mechanical properties of fish bones, It is necessary to focus on deriving benchmarking elements for application.

Author Response

thank you so much for your comments and suggestions.

Reviewer 2 Report

In the manuscript-“Micro-architectural investigation of teleost fish rib inducing pliant mechanical property”, the authors try to establish guiding principles toward controlling the flexibility of regenerated bone through the analysis of the composition, microstructure, histology, and mechanical properties of teleost fish ribs. The paper provides the reader with some important design criteria for engineering tough and soft type of bone. However, there are some minor issues that need to be fixed prior to publication.  

Broad comments-

  • Please include the significance of stacked collagen layers in the discussion section. What could be the consequences of such a characteristic appearance?

  • At several locations, authors refer to the stage of development. As per my understanding, the studies are not pertaining to different stages of bone development and in the manuscript only a single stage i.e., adult has been considered. Please revise such statements accordingly e.g. line 18-19 - because understanding the structural and functional change of bone during its development would be valuable. Also, they call Rib 1 more mature as compared to the other two. Please elaborate on this point- why aren’t the other two ribs fully matured in adult fish?

  • Please ensure that wherever there are statements based on previously established facts, appropriate reference is provided.
    • g. Line 103- Fish bones have soft and strong mechanical properties.
    • Line 182-183- regeneration of bone defects of small size, such as a tiny alveolar bone defect in periodontal diseases, has been almost achieved.
    • Line 190-191- Rib1 of Carassius langsdorfii is a more mature bone than Rib5, and Rib5 is a more mature bone than Rib9.

  • The authors are advised to include a short explanatory title for all the figures.

  • Define acronyms HE, EDTA at their first appearance.

  • This suggestion is related to text formatting- please ensure that the figure and its legend are on the same page- fig 3, fig 5

  • Specific comments-

  • A number of relevant and key publications are missing. The authors are recommended to do a more thorough literature search and cite relevant publications. Few are listed below-

  1. Koons, Gerry L., Mani Diba, and Antonios G. Mikos. "Materials design for bone-tissue engineering." Nature Reviews Materials (2020): 1-20.
  2. Fiedler, I. A. K., et al. "Microstructure, mineral and mechanical properties of teleost intermuscular bones." Journal of Biomechanics 94 (2019): 59-66.
  3. Cohen, Liat, et al. "Comparison of structural, architectural and mechanical aspects of cellular and acellular bone in two teleost fish." Journal of Experimental Biology 215.11 (2012): 1983-1993.

  • Line 98- The statement should be changed to ‘the data are represented as mean + SD’

  • It is not clear- which rib’s data i.e., 1 or 5 or 9 is presented in Fig 1 E.

  • Fig 2- Please add standard deviations to the bars representing water, organic and inorganic content. In my opinion water content is about 5-7% and organic content is about 35-40%. Also, this % data is repetitive as it appears in the figure caption as well as in results text.

  • Fig 2- please enlarge- labels are not easy to read; especially for part A- figure captions are not readable.

  • Fig 3- scale bars are not readable for B to G.

  • Fig 4- B and C- consider enhancing the contrast and increasing the font size of the scale bars. Also, please include the information- which rib is imaged in figure parts A, B, and C for better understanding.

  • Line 152, 153- osteocytes existing on the outside of the bone matrix showed higher orientation than cells in the inner region. Here, the sentence construction sounds ambiguous especially the term ‘ higher orientation’. Do you think the term ‘better cellular alignment’ would make more sense?

  • The manuscript needs some language editing for better readability. E.g

  1. Line 29 – decade—change to –decades
  2. Line 57- effect of waters—change to--the effect of water content
  3. Line 58- were heated with different temperatures--change to--at different temperatures
  4. Line 184- complicating—change to --complicated
  5. Line 195- characteristic –change to—characteristically

Author Response

(The authors gave the same response as above.)

Reviewer 3 Report

  1. This is an interesting paper that studies the mechanical properties of the teleost fish rib. It would be good if the authors can also provide detailed analysis of the material composition which confers such amazing performance of the fish rib. Evaluation of the material composition would be critical towards the understanding of the pliant property in the fish rib.
    1. Is it possible to break down and identify the specific composition of organic and inorganic mass?

  1. Besides the different types of materials used for bone tissue engineering, it would be good if the authors also provide more discussion on the fabrication techniques for bone scaffolds in the Introduction with relevant references.
    1. Solvent casting and particulate leaching
      1. "Combined porogen leaching and emulsion templating to produce bone tissue engineering scaffolds." International Journal of Bioprinting6, no. 2 (2020): 99-113.
    2. Freeze-drying
      1. "Fabrication and in-vitro biocompatibility of freeze-dried CTS-nHA and CTS-nBG scaffolds for bone regeneration applications." International Journal of Biological Macromolecules149 (2020): 1-10.
    3. Electrospinning
      1. "Engineered dual-scale poly (ε-caprolactone) scaffolds using 3D printing and rotational electrospinning for bone tissue regeneration." Additive Manufacturing36 (2020): 101452.
    4. 3D printing
      1. "Print me an organ! Why we are not there yet." Progress in Polymer Science97 (2019): 101145.

  1. Another concern is the use of freeze-drying process for microstructure observation. This would likely distort the microstructure of water-containing fish bones (~10%). Would it be better to use critical point drying technique to preserve the microstructure of fish bones?

  1. How many samples are used for the mechanical testing (Stress vs strain, young modulus)?

Author Response

(The authors gave the same response as above.)

Round 2

Reviewer 1 Report

Some of the contents have been revised, but I still think that the following information needs to be supplemented.  

In 3.1, it is very difficult to measure the mechanical properties of fish bones and requires advanced technology, but details for measure method are missing.

In the discussion, there are many references to existing studies, but in comparison with this, it is necessary to mention and review main factors for maintaining flexible bone tissue properties.

In the conclusion, the logic for reaching the conclusion is very insufficient. In the case of this paper, it focuses on investigating structure and measuring mechanical properties the introduction, but for this, creativity and originality in measurement technology are required. If it is intended to be applied to other parts by measuring the mechanical properties of fish bones, It is necessary to focus on deriving benchmarking elements for application.

Author Response

Thank you for taking the time to review our paper.

The parts corrected this time are shown in blue in the manuscript.

We have also revised the English description again throughout the manuscript.

Please check in the attached file.

Reviewer 3 Report

The replies to the comments are satisfactory. The manuscript can be accepted in present form.

Author Response

Thank you for taking the time to review our paper.

Minor spell check of English language was carefully checked and revised again throughout the manuscript.